# High-Temperature-Treated LTX Zeolites as Heterogeneous Catalysts for the Hock Cleavage

**Jan Drönner [1], Karim Bijerch [1], Peter Hausoul [2], Regina Palkovits [2] and Matthias Eisenacher [1,*]**

1 Circular Transformation Lab Cologne, TH Köln-University of Applied Sciences, 51379 Leverkusen, Germany
2 Institut für Technische und Makromolekulare Chemie, RWTH Aachen University, 52074 Aachen, Germany
* Correspondence: matthias.eisenacher@th-koeln.de

**Abstract:** Hydroxybenzene, commonly known as phenol, is one of the most important organic commodity chemicals. To produce phenol, the cumene process is the most used process worldwide. A crucial step in this process is the Hock rearrangement, which has a major impact on the overall cumene consumption rate and determines the safety level of the process. The most used catalyst for the cleavage of cumene hydroperoxide (CHP) is sulfuric acid. Besides its strong corrosive property, which increases plant investment costs, it also requires neutralization after the decomposition step to prevent side reactions. In this study, we show that high-temperature-treated Linde Type X (LTX) zeolites exhibit a high activity for the peroxide cleavage step. In addition, the structure–activity relationship responsible for this good performance in the reaction system of the HOCK rearrangement was investigated. XRPD analyses revealed the formation of a new phase after temperature treatment above 900 °C. The Si/Al ratio determined by EDX suggested the formation of extra-framework aluminum, which was confirmed by solid-state NMR analysis. The newly formed extra-framework aluminum was found to be responsible for the high catalytic activity. BET analyses showed that the surface area drops at higher calcination temperatures. This leads to a lower catalytic activity for most known reactions. However, for this study, no decrease in activity has been observed. The newfound material shows extraordinarily high activity as a catalyst in the HOCK cleavage and has the potential to be a heterogeneous alternative to sulfuric acid for this reaction.

**Keywords:** Hock cleavage; heterogeneous catalysis; cumene hydroperoxide; extra-framework aluminum; zeolites

## 1. Introduction

Phenol is one of the most important organic commodity chemicals; with $12.7 \times 10^6$ t of phenol and $7.8 \times 10^6$ t of acetone produced annually worldwide, the cumene process is still by far the most used process today [1,2]. It was discovered and developed at the same time in the USSR and Germany. In the years 1943–1946, Udris and Sergeev developed a method for the synthesis of phenol and acetone via the acidic decomposition of cumene hydroperoxide (CHP). In 1944, Hock and Lang published a three-step route from benzene and propylene to cumene via alkylation and following autoxidation to CHP, that is cleaved to phenol and acetone via acidic decomposition (Scheme 1) [3–5]. The cleavage step is named 'Hock rearrangement' and can be catalyzed by Brønsted or Lewis acids [6]. The Hock rearrangement has a major impact on the overall cumene consumption rate of the entire phenol plant [7–10]. The industrially employed cumene process starts with the alkylation of benzene with propylene to cumene, followed by the oxidation of cumene to CHP in the presence of air and free radical initiators. Once a free radical initiator is added, CHP will catalyze the oxidation reaction [11]. In contrast to typical radical reactions of peroxides (induced by light, heating, or other methods), the cleavage of CHP is a rearrangement of hydroperoxides usually carried out as a sulfuric-acid-catalyzed reaction. The cleavage reaction starts with the protonation of the peroxide, followed by the elimination of water

due to the migration of the phenyl group to the peroxide oxygen atom. This forms a carbocation, which is attacked by a water molecule, and one proton switches to the oxygen of the phenyl group it migrated to earlier. This oxonium ion finally decomposes into acetone and phenol [6] (Scheme 1).

**Scheme 1.** General mechanism of the cumene process.

In the industry, sulfuric acid is used as a cheap and good-working catalyst. Unfortunately, the cheap price of this mineral acid faces different disadvantages. One is the need to add a neutralizing agent after the decomposition step to prevent side reactions and to remove the aggressive acid before rectification. The salts formed by neutralization must be removed to prevent contamination and blockage of the evaporator of the rectification column [12]. The necessity of corrosion-resistant material and the required neutralization and salt removal are accompanied by the relatively low price for sulfuric acid. After commercialization in Europe and the USA after World War II, various other homogeneous catalysts were investigated [13–19], but none of them showed superior performance to sulfuric acid in terms of catalytic and cost efficiency. To solve the persistent problems with sulfuric acid being a corrosive and homogeneous catalyst, various approaches to replace the homogeneous catalyst with a solid catalyst have been studied in the past 20 years [20]. Particularly regarding their low price, modifiability, and—due to their strong acidity—capability to replace several mineral acids, zeolites in particular have been tested, including USY-, Beta-, and ZSM-5-type zeolites [21–27].

Faujasite zeolites have been found to be active as catalysts for the hydroperoxide cleavage; however, mainly Y-type zeolites have been used up to now [26]. Therefore, X-type zeolites have been studied in the present work.

## 2. Results and Discussion

### 2.1. XRPD

The XRPD patterns of the calcined LTX zeolites are shown in Table 1. After calcination at temperatures between 400 and 700 °C, the crystalline structure does not change and matches the pattern of LTX (COD 9005387). The degree of crystallinity (DOC) was calculated by subtracting the instrumental background from the sample's diffraction pattern, re-determining the background of the measurement, calculating the area below the background curve, and comparing it to the area of the whole pattern of the investigated catalyst. Although the pattern does not change, the crystallinity of the zeolite decreases with higher calcination temperatures. After 2 h at 800 °C, the LTX structure mostly collapses and leads to an amorphous structure. After calcination at 900 °C and especially after calcination at 1000 °C, the structure changes thoroughly to a different crystal structure with high crystallinity (Figures 1 and 2).

The reflexes of these patterns could be assigned to three different phases. Above a calcination temperature of 800 °C, but under 1000 °C, kumdykolite ($NaAlSi_3O_8$) is formed, which is an orthorhombic polymorph of albite [28]. The LTX zeolite calcined at 1000 °C shows the pattern of $Al_4Na_{3.49}O_{16}Si_4$ nepheline [29] and cubic $Al_2O$ [30]. The formation of $Al_2O$ shows that with high-temperature treatment, extra-framework aluminum in a formerly LTX zeolite could be created.

**Table 1.** Crystallinity, Si/Al ratio, surface area, and median pore width of the synthesized LTX catalyst after calcination at different temperatures.

| Calcination Temperature /°C | Crystallinity /% | Si/Al Ratio | Surface Area /$m^2g^{-1}$ | Median Pore Width /nm |
|---|---|---|---|---|
| 0 | 78 | 1.24 | 417 | 0.92 |
| 400 | 78 | 1.26 | 417 | 0.92 |
| 500 | 78 | 1.26 | 291 | 0.80 |
| 600 | 69 | 1.25 | 19 | 0.58 |
| 700 | 52 | 1.31 | 7 | 0.58 |
| 800 | 20 | 1.18 | 1.4 | 0.77 |
| 900 | 87 | 1.27 | 1.3 | 1.20 |
| 1000 | 95 | 0.87 | 0.1 | 0.1 |

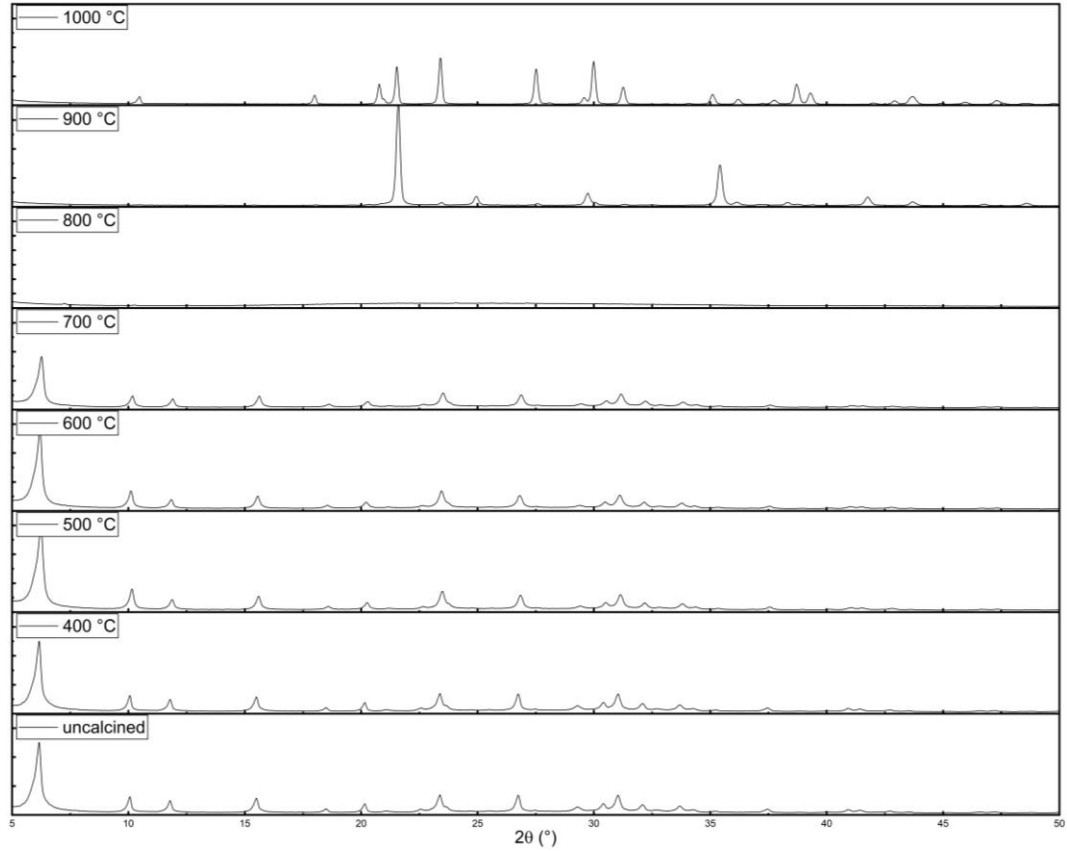

**Figure 1.** XRPD patterns of LTX zeolite after calcination at different temperatures.

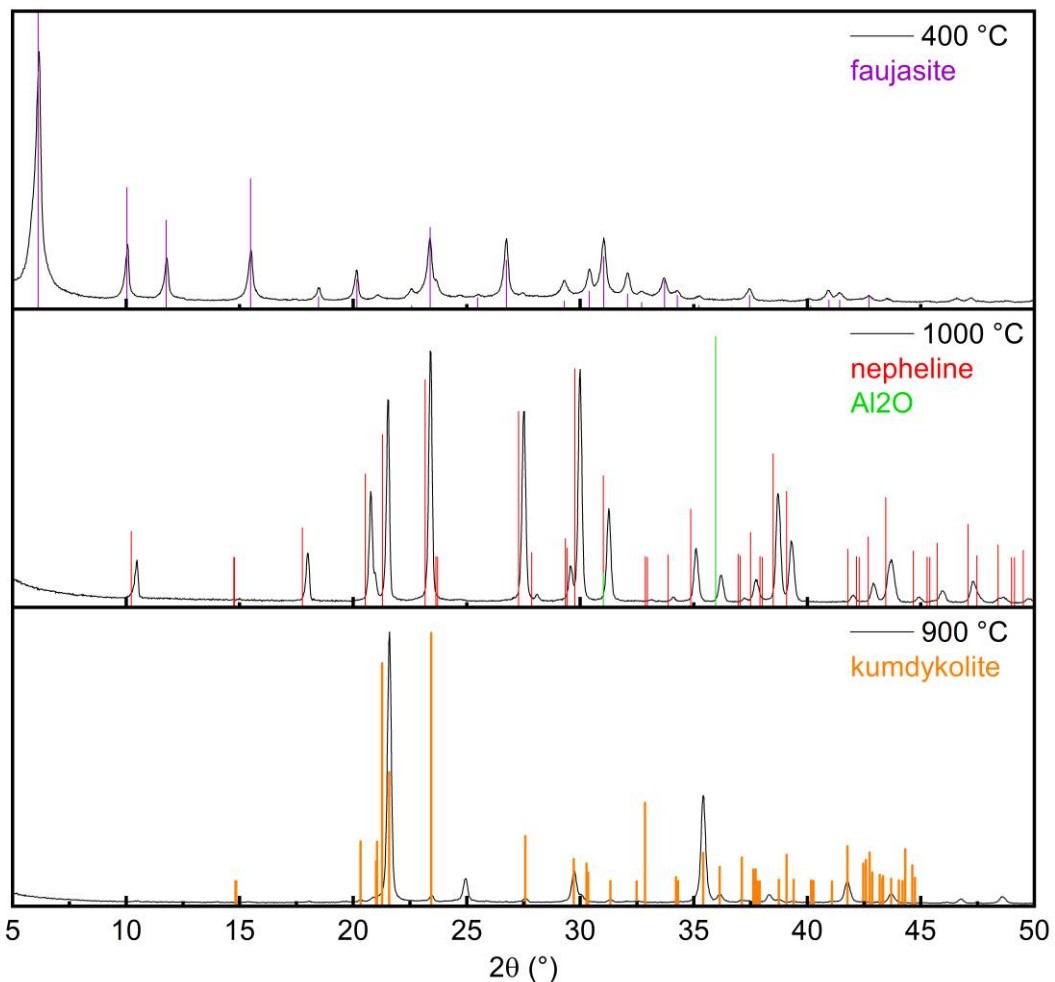

**Figure 2.** XRPD patterns of the LTX zeolite calcinated at 400 °C and a faujasite pattern as reference (**top**), the newly formed kumdykolite structure after high-temperature treatment at 900 °C (**middle**), and the nepheline structure (reference in red) with overlapping reflexes from $Al_2O$ (green) as an extra-aluminum framework after calcination at 1000 °C (**bottom**).

### 2.2. SEM-EDX

The acquired EDX data are based on SEM. The structural change is also seen by a change in the Si/Al ratio. Up to a calcination temperature of 600 °C, the Si/Al ratio stays nearly unchanged between 1.24 and 1.26. At 700 °C calcination temperature, the Si/Al ratio begins to change and eventually decreases to under 1 after high-temperature treatment at 1000 °C, meaning that it would no longer be a zeolite by definition (Table 1) [31]. The low Si/Al ratio can be explained with the relatively little penetration depth of the method (about 5–10 μm). The formation of an extra-aluminum framework on the catalyst's surface if treated by high temperatures above 900 °C is assumed. This framework is formed on the surface of the zeolite particles, which leads to a lower Si/Al ratio of the treated zeolite.

### 2.3. BET

With a higher calcination temperature, the surface area of the catalyst is reduced as expected. Untreated, the zeolite has a surface area of 417 $m^2g^{-1}$, and after heat treatment at 500 °C the surface area drops to 291 $m^2g^{-1}$. Above 800 °C, the catalyst changes from a crystalline structure to an amorphous state and, with higher temperatures, changes back to a new crystalline structure; the surface area decreases to 19 $m^2g^{-1}$ after treatment at 800 °C and drops even further, reaching 0.05 $m^2g^{-1}$ after calcination at 1000 °C. The median pore width of the newly formed structure, however, also becomes lower with increasing

calcination temperature, from 0.92 nm without heat treatment to 0.58 nm after calcination at 800 °C, but the newly formed crystalline structure after heat treatment above 800 °C shows a rising median pore width, with even bigger pores than the initial zeolite (1.20 nm at 950 °C) (Table 1). As a consequence of the collapse of the surface area and the pores, the characterization of the acidity by $NH_3$-TPD did not lead to any evaluable results. The isotherms and summary reports of the BET experiments are shown in Figure S3–S16 in the supplementary file.

*2.4. ssNMR*

Figure 3 shows the solid-state $^{27}$Al NMR spectra of the zeolites calcined at 400 °C and 1000 °C. The spectrum of the calcined zeolite at 400 °C consists only of one signal (at 63.10 ppm) which is related to $Al^{IV}$ clusters [32]. The spectrum of the high-temperature-treated (1000 °C) zeolite shows signals of cluster $Al^{IV}$ at 61.90 ppm and additional framework aluminum atoms in the vicinity of extra-framework aluminum at 66.05 ppm [32]. The change in the shift of the cluster $Al^{IV}$ is caused by the change in the crystal structure.

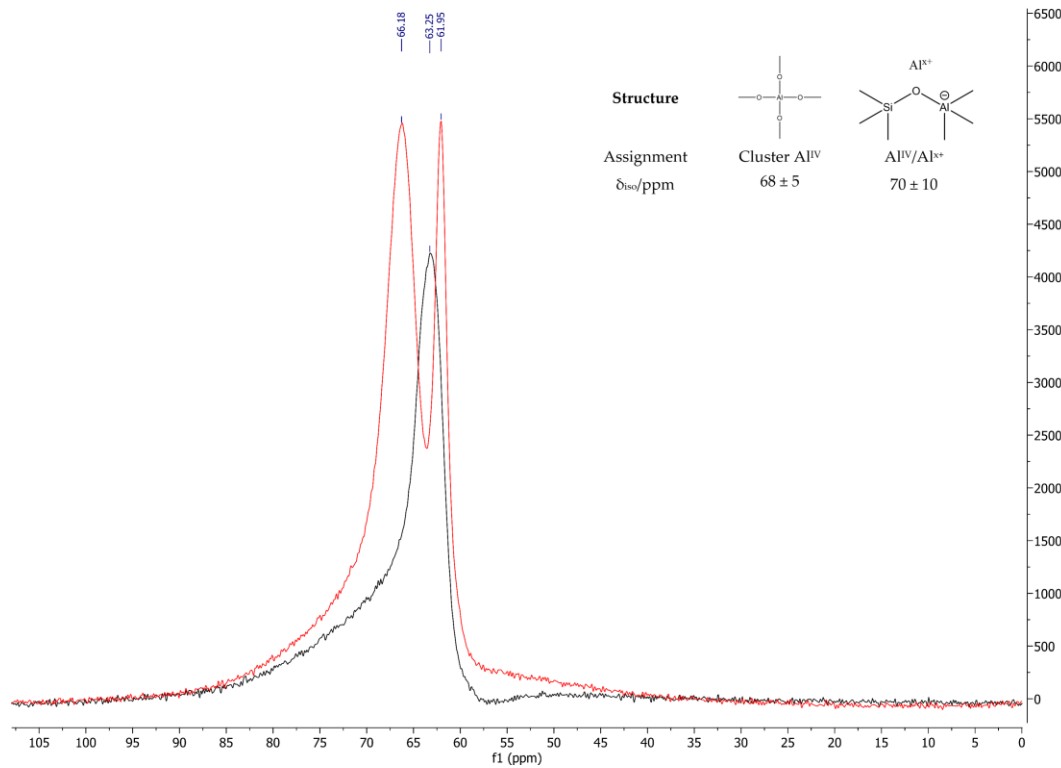

**Figure 3.** Solid-state NMR spectra of the modified catalyst calcined at 400 °C (black) and calcined at 1000 °C (red).

*2.5. Cleavage of CHP*

Figure 4 shows the conversion of CHP (X (CHP)), selectivity towards phenol (S (phenol)), and the overall yield of phenol (Y (phenol)) obtained under the same reaction conditions using the catalysts calcined at different temperatures. Conversion, yield and selectivity were estimated based on GC analysis of the product mixture. An exemplary chromatogram and spectrum of the product mixture of one of the experiments is shown in Figure S1 in the supplementary file. Figure S2 in the supplementary file shows the reference spectrum of phenol and the temperature profile of the used GC method.

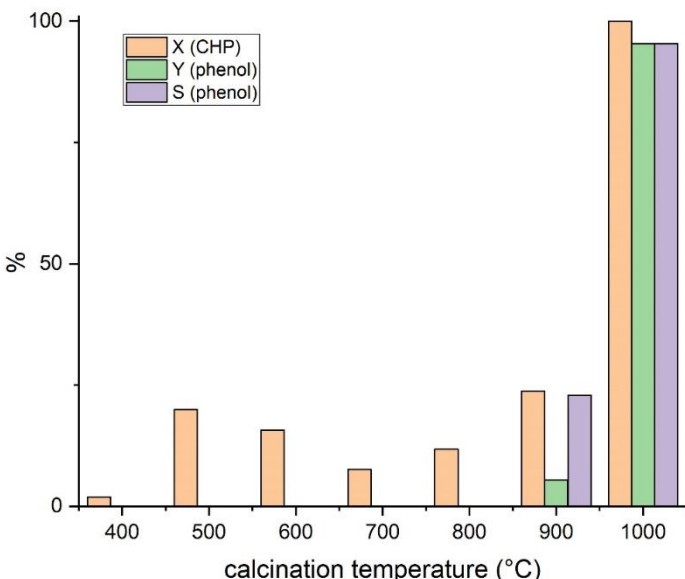

**Figure 4.** Yield of phenol in the Hock rearrangement of CHP in chloroform with different catalysts; 45 bar, 80 °C, 8.62 mL substrate.

All catalysts with the LTX structure (calcined from 400 °C to 800 °C) show a poor conversion of CHP of 5 to 20%. However, these reactions did not lead to any formation of phenol. Acetophenone was observed as the only reaction product. With the catalyst calcined at 1000 °C, the conversion of CHP was 100% with 94% selectivity towards the desired reaction product phenol.

Kinetic studies show that the cleavage of CHP takes place within just a few minutes (Figure 5). The reaction can be run at reaction rates which are comparable with the reaction rates of the sulfuric-acid-catalyzed decomposition of CHP [33].

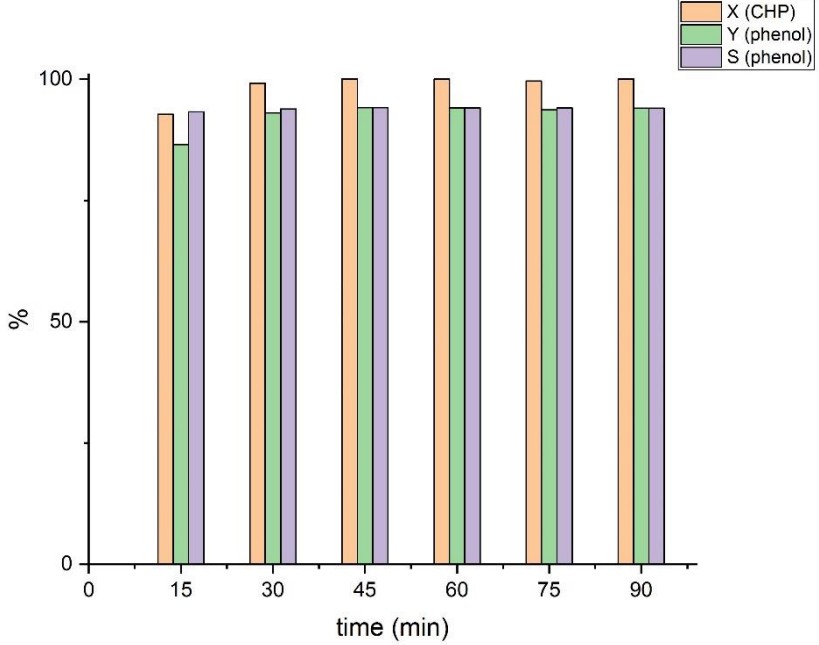

**Figure 5.** Conversion rate of CHP, yield, and selectivity to phenol using the LTX zeolite calcined at 1000 °C over time; 8.62 mL substrate in 50 mL chloroform, 1 g catalyst, 45 bar, 80 °C, 300 rpm in 75 mL autoclaves.

## 3. Experimental

### 3.1. Materials

Sodium silicate solution (27.6% $SiO_2$, 8.5% $Na_2O$ in deionized water) and sodium aluminate (50%) were purchased from VWR Chemicals. Sodium hydroxide ($\geq$99.4%) and chloroform ($\geq$99.5%) were obtained from Fisher Chemical. CHP (80% in cumene) was purchased from Sigma-Aldrich.

### 3.2. Preparation of LTX Zeolites

LTX catalysts were prepared using two solutions (A, B). Bottles made of polyethylene were used to prevent contamination with foreign metal ions. Solution A: Sodium hydroxide (100 g, 2.5 mol) was dissolved in deionized water (100 g). Sodium aluminate (127.5 g, 0.78 mol) was added and the solution was heated to 100 °C and stirred until completely dissolved. After letting the mixture cool to 25 °C, water (202.5 g) was added. Solution B: solution A (50 g) was mixed with water (306 g) and sodium hydroxide (29.56 g, 0.739 mol). Solution C: sodium silicate solution (108.85 g), distilled water (306 g), and sodium hydroxide (29.56 g, 0.739 mol) were mixed until dissolved. Solutions B and C were combined quickly and stirred for about 1 h at 700 rpm. Subsequently, this mixture was held at 90 °C without any agitation for 24 h. The precipitated zeolite was filtered off via vacuum distillation, washed with water to pH < 10, and dried at 100 °C. The zeolite (21 g) was then calcined for 2 h at temperatures ranging from 400 °C to 1000 °C.

### 3.3. Characterization of Catalysts

X-ray powder diffraction (XRPD) patterns were recorded using a D2 PHASER diffractometer from BRUKER AXS GMBH, with Cu-K$_\alpha$ radiation with a wavelength of 1.54060 Å in the range 2$\Theta$ = 5–50° with 0.05° of step size.

Energy dispersive X-ray spectroscopy (EDX) was carried out using a CamScan 24 SEM, with an (EDX) Oxford Si(Li) SATW detector. The software used was INCA Energy 250.

The BET surface area was measured using a Micromeritics ASAP 2060 surface area analyzer. To calculate the surface area, the BET equation was used, while pore volume and pore size were obtained using the pore size determination method [34]

The $^{27}Al$ solid-state magic angle spinning NMR spectra of the calcined zeolites were recorded at room temperature via direct polarization with a Bruker Avance III HD 500 MHz spectrometer using a 4 mm rotor and a spinning frequency of 10,000 kHz.

### 3.4. Experimental Setup for the Cleavage of CHP

For the chemocatalytic cleavage of CHP, a 5000 Multi Reactor Heater system from Parr was used. The reaction was carried out in 75 mL autoclaves with 1 g of each catalyst. Then, CHP solution (10 mL, 80% in cumene) and chloroform (50 mL) were added. The autoclave was heated to constant 80 °C and stirred with 300 rpm. After 120 min, the autoclave was cooled rapidly, and a sample was taken. The zeolite was separated using a syringe filter. Reaction mixtures were analyzed via gas chromatography using a Shimadzu GCMS-QP2010 SE equipped with a RTX-200MS column.

## 4. Conclusions

The calcination study of LTX catalysts shows that the crystal structure of the zeolites begins to collapse at about 800 °C and a material with low crystallinity is formed. At temperatures above 900 °C, nepheline as a new crystal structure with an $Al_2O$ extra-framework is formed.

The extra-framework aluminum species, which was formed at calcination temperatures above 900 °C, could clearly be identified via EDX and ssNMR studies. This leads to an impoverishment of aluminum in the crystal lattice of this material which is, according to Loewenstein's rule, not a zeolite anymore [31].

BET analyses demonstrated that all catalysts calcined at high temperatures nearly lose their surface areas entirely. Since the catalyst calcined at 1000 °C showed the highest

performance for the cleavage of CHP, it can be assumed that there is no correlation between surface area and catalytic performance.

The catalyst calcined at 1000 °C shows a superior catalytic performance leading to a yield of 94% towards phenol. Kinetic studies showed that this yield can be achieved within reaction times of just a few minutes.

This result shows that the yield and the reaction rate for this reaction are comparable to the yield obtained using sulfuric acid as a catalyst [34]. However, the catalyst presented overcomes all the aforementioned intrinsic disadvantages of sulfuric acid.

Future work will focus on optimizing operating parameters, determining detailed kinetic data, and on studying the long-term stability of the catalyst. Furthermore, the Si/Al ratio will be investigated via ICP and XRF. The morphology and acidity of the newly formed structure will be studied extensively with additional characterization techniques such as FTIR, TGA, UV/VIS, and 2D ssNMR.

Since the active catalyst calcined at more than 900 °C does not show the LTX structure anymore, future work will also focus on looking for a cheaper manufacturing process for the active species not starting from LTX.

**Supplementary Materials:** The following supporting information can be downloaded at: https://www.mdpi.com/article/10.3390/catal13010202/s1, Figure S1: Exemplary chromatogram and spectrum of the product mixture of one of the experiments. The spectrum shown is that of phenol; Figure S2. Left: Reference spectrum of phenol; Right: Temperature profile of the used GCMS-method; Figure S3. Report of BET analysis of a not calcined LTX zeolite; Figure S4. Linear plotted isotherm of a not calcined LTX zeolite; Figure S5. Report of BET analysis of a LTX zeolite calcined at 500 °C; Figure S6. Linear plotted isotherm of a LTX zeolite calcined at 500 °C; Figure S7. Report of BET analysis of a LTX zeolite calcined at 600 °C; Figure S8. Linear plotted isotherm of a LTX zeolite calcined at 600 °C; Figure S9. Report of BET analysis of a LTX zeolite calcined at 700 °C; Figure S10. Linear plotted isotherm of a LTX zeolite calcined at 700 °C; Figure S11. Report of BET analysis of a LTX zeolite calcined at 800 °C; Figure S12. Linear plotted isotherm of a LTX zeolite calcined at 800 °C; Figure S13. Report of BET analysis of a LTX zeolite calcined at 900 °C.; Figure S14. Linear plotted isotherm of a LTX zeolite calcined at 900 °C; Figure S16. Linear plotted isotherm of a LTX zeolite calcined at 1000 °C.

**Author Contributions:** Conceptualization, J.D. and M.E.; methodology, J.D.; validation, P.H., R.P., and M.E.; formal analysis, P.H. and M.E.; investigation, J.D. and K.B.; data curation, J.D.; writing—original draft preparation, J.D.; writing—review and editing, P.H. and M.E.; visualization, J.D.; supervision, M.E.; project administration, R.P. and M.E.; funding acquisition, M.E.. All authors have read and agreed to the published version of the manuscript.

**Funding:** This study was funded by the German Federal Ministry of Education and Research (FKZ: 031B0671).

**Data Availability Statement:** Not applicable.

**Acknowledgments:** The authors are grateful to Gerd Braun (TH Köln—University of Applied Sciences) for carrying out the EDX analyses and to Noah Avraham (RWTH Aachen University) for conducting the BET analyses.

**Conflicts of Interest:** The authors declare no conflict of interest.

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
