# Peer review of "High-Temperature-Treated LTX Zeolites as Heterogeneous Catalysts for the Hock Cleavage"

_catalysts, doi:10.3390/catal13010202_

Round 1
Reviewer 1 Report
Eisenacher et al reported that the treatment of LTX zeolites above 900 °C can form a new phase (Al4Na3.49O16Si4 nepheline and cubic Al2O), which exhibited a high activity for the Hock cleavage. The results are interesting and innovative, however the manuscript needs major revision before considering being accepted by Catalysts. The comments are the following:
1. Since Hock rearrangement is a proton acid catalyzed reaction, the acidity of the catalyst should be the primary concern. However, the acidity of the catalyst has not been characterized in this paper, which cannot be ignored. The author must add the acid characterization data for the catalysts and make some correlation analysis with the catalytic reaction results.
2. At the reaction temperature of 80 °C, the strong-acid cation exchange resin catalyst for proton acid catalytic reaction seem to have more advantages. Its -SO3H density is generally higher than molecular sieve, with stronger acidity and the properties closer to sulfuric acid catalyst, and the catalyst is cheaper. Thus, the comparison between the strong-acid cation exchange resin catalyst and molecular sieve catalyst for this reaction should be given in the paper (at least some references should be cited for comparison and comment in the Introduction section).
3. From the results of the paper, the structure of LTX zeolites was completely changed after calcination at 1000℃, and all channels collapsed (surface area of 0.05m2/g). Thus, it can be concluded that the new obtained catalyst after calcined at 1000℃ is more concerned with its crystal structure and composition, but not the pore system. So, what is the significance of using LTX zeolites as precursor? After all, the preparation process of molecular sieve is tedious and the molecular sieve is expensive. Is it possible that the Al4Na3.49O16Si4 nephenine and cubic Al2O phase can be directly synthesized from conventional amorphous aluminosilicate precursor with a certain proportion, which can greatly reduce the catalyst preparation cost?
Reviewer 2 Report
This manuscript released a high temperature treated LTX zeolite on enhancing cleavage of cumene hydroperoxide to replace sulfuric acid. pXRD, EDX, NMR, and BET analyses were conducted to analyze its structure, composition and morphology. It shows that the calcination temperature is critical to the catalytic activity. There is no doubt that this manuscript could provide potential to be a alternative catalyst for this reaction. There are several sites need to be modified.
1. In the abstract, the full name should be give for LTX.
2. Why did the author choose LTX zeolites for investigation. What’s the advantage compared to other zeolites listed in the introduction part? Should summarize and document the previous studies of these zeolites in details for cumene hydroperoxide cleavage.
3. The XRD pattern showed in Fig 1and 2 are not clear to read. Please re-plot using origin or sigma-plot software.
4. Is EDX data based on the SEM or TEM? Please state that clearly. Also, the author should provide other techniques to quantify the Si/Al ratio, such as ICP-OES or ICP-MS.
5. The N2 adsorption/desorption isotherms were missing. Also, the GC data were missing to determine the product.
6. Other characterization techniques should be provided to check the morphology of catalysts with different calcination temperature.
7. How about the TOF? It seems that the surface area dropped dramatically over 800 C though the catalytic activities were higher compared to those of low temperature annealed catalysts.
Round 2
Reviewer 1 Report
Although there are still some scientific questions that have not been thoroughly explored, it can be accepted as the Type “Communication”.
Reviewer 2 Report
The revised manuscript addressed most of my comments and I agree to publish at current stage.